# Efficacy and safety of berberine plus 5-ASA for ulcerative colitis: A systematic review and meta-analysis

Jilei Li[1☯], Chenchen Zhang[2], Yanchao Xu[1], Lili Yang [3☯]*

1 Department of Oncology Diseases, Henan Province Hospital of Traditional Chinese Medicine, Zhengzhou, Henan, China, 2 Graduate School Department, Beijing University of Chinese Medicine, Beijing, China, 3 Henan Province Hospital of Traditional Chinese Medicine, Zhengzhou, Henan, China

☯ These authors contributed equally to this work.
* 15670259809@163.com

**Data Availability Statement:** All relevant data are within the manuscript and its Supporting Information files.

**Funding:** The author(s) received no specific funding for this work.

## Abstract

Purpose: This study aimed to assess the efficacy and safety of berberine(BBR) plus 5-aminosalicylic acid (5-ASA) for treating ulcerative colitis (UC). Methods: A comprehensive search was conducted in electronic databases, including Medline/PubMed, Sinomed, Embase, CNKI, Wanfang, and VIP, through January 2024 to identify all randomized controlled trials (RCTs) that administered BBR conjunction in standard therapy(5-ASA) for to support the treatment of UC. The data were synthesized using a meta-analysis approach with RevMan 5.4.1. The primary endpoint was the clinical efficacy rate. In contrast, the secondary endpoints included the Baron score, disease activity index (DAI) score, symptom relief latency, inflammatory markers, immunological indicators, and adverse events. Results: In this analysis, 10 RCTs comprising 952 patients with UC were examined. BBR considerably improved the clinical efficacy rate (RR = 1.22, 95% CI [1.15, 1.30], $P <$ 0.00001), attenuated the Baron score (SMD = -1.72, 95% CI [-2.30, -1.13], $P < 0.00001$) and reduced the DAI score (SMD = -2.93, 95% CI [-4.42, -1.43], $P < 0.00001$). Additionally, it ameliorated clinical symptoms (SMD = -2.74, 95% CI [-3.45, 2.02], $P < 0.00001$), diminished inflammatory responses (SMD = -1.59, 95% CI [-2.14, 1.04], $P < 0.00001$), and modulated immune reactions (SMD = 1.06, 95% CI [0.24, 1.87], $P < 0.00001$). Nonetheless, the impact of BBR on reducing adverse reactions was not statistically significant (RR = 0.75, 95% CI [0.42, 1.33], $P > 0.05$). Conclusion: BBR demonstrates substantial efficacy in treating UC without causing severe adverse reactions and may serve as a viable complementary therapy. However, its clinical application warrants confirmation by additional high-quality, low-bias RCTs.

## Introduction

Ulcerative colitis (UC) is an inflammatory bowel disease primarily manifesting as abdominal pain, diarrhea, and purulent, mucoid stools [1]. The etiology of UC is complex, predominantly

**Competing interests:** The authors have declared that no competing interests exist.

affecting the rectum and sigmoid colon, with recurrent flare-ups and difficulty in achieving complete remission [2]. Globally incidence rates of UC are increasing; North America has reported prevalence rates between 139.8 and 286.3 per 100,000 individuals, while China has observed an incidence of approximately 11.6 per 100,000 individuals, indicating that UC is a substantial public health concern acknowledged by the World Health Organization (WHO) [3]. Regarding treatment for UC, mesalamine and sulfasalazine are integral to the 5-ASA medication class and constitute the cornerstone of clinical therapy for mild to moderate UC [4]. Progress in medical science has unveiled the effectiveness of immunosuppressive and corticosteroid drugs, with recent introductions of biological agents such as ustekinumab bringing forth novel therapeutic avenues [5]. Despite their benefits, these conventional treatments pose considerable challenges, including suboptimal mucosal healing, prohibitive costs, and a spectrum of adverse effects [6, 7], necessitating novel drug development for UC management. Current research posits that potent, nontoxic natural substances may be viable adjunctive treatments for standard UC therapy [8]. BBR, a potent alkaloid from Coptis chinensis and Phellodendron amurense, is noted for easing abdominal discomfort and diarrhea in UC treatment [9]. Empirical evidence from traditional Chinese medicine suggests that formulations including Coptis and Phellodendron lessen inflammatory responses, reducing symptomatology [10]. A meta-analysis of animal studies indicated that BBR reduces the DAI score and histological colitis score in UC animal models [11]. Mechanistically, BBR may exert its effects by decreasing myeloperoxidase (MPO) activity and malondialdehyde (MDA) levels, reducing the expression of pro-inflammatory factors such as interleukin-1β (IL-1β), interleukin 6 (IL-6), and tumor necrosis factor α (TNF-α), and increasing the levels of the tight junction proteins zonula occludens-1 (ZO-1) and occludin. However, this animal study evaluated the efficacy of BBR, which does not directly guide its clinical application. As a commonly used adjunctive medication, the effectiveness of BBR combined with 5-ASA treatment for UC is still uncertain, and there is currently a lack of corresponding clinical research. What makes us even more curious is that Xu 2020 [12] believes that BBR is ineffective in improving diarrhea and bloody purulent stools in UC patients, which contradicts most of the existing research findings. Therefore, it is necessary to conduct further research to explore the efficacy of BBR combined with 5-ASA in treating UC. Our study aimed to evaluate the effects of BBR as an adjunct treatment on the efficacy and safety of UC.

## Materials and methods

### Search strategy

The methodology for this study was aligned with the Cochrane Handbook, and the reporting conformed to the PRISMA 2020 checklist [13] (S1 Checklist). This systematic review has been appropriately registered with the PROSPERO platform and assigned the identifier CRD42022376150. A comprehensive literature search was conducted across databases, including Medline/PubMed, Sinomed, Embase, CNKI, Wanfang, and VIP, imposing no linguistic constraints and spanning from their inception through January 2024. For the retrieval of literature, the search terms used were "ulcerative colitis", "ulcerative", "inflammatory bowel disease", "BBR", "BBR hydrochloride", "umbellate", "randomized" and "randomized controlled trial". All studies identified in the literature search are listed in S1 Table. The raw extracted data from the 10 included studies can be found in S2 Table.

The sequence used in the PubMed database was as follows:

#1: "ulcerative colitis" OR "ulcerative" OR "inflammatory bowel disease" [Mesh]

#2: "berberine" OR "berberine hydrochloride" OR "umbellatine" [Title/Abstract]

#3: #1 AND #2

#4: "randomized" OR "randomized controlled trial" [Title/Abstract]

#5: #3 AND #4

## Inclusion criteria

(1) Study Participants: This study included individuals aged 18 years or older with a clinical diagnosis of UC according to the most recent diagnostic guidelines [14]. The inclusion criteria for Randomized Controlled Trials (RCTs) were as follows: (2) Intervention: The control group received standard 5-ASA treatment, while the treatment group was treated with a combination of 5-ASA and BBR. (3) Outcome measures include symptom relief (clinical efficacy rate); serological indicators (baron score, DAI assessment, symptom amelioration analysis, inflammatory cytokines, immune response); and safety profiles (adverse reactions).

## Exclusion criteria

The following are the reasons for exclusion from this study: (1) were not Randomized Controlled Trials (RCTs), publications that have been duplicated, or articles with statistical inaccuracies (Original data not mentioned or statistical methods not referenced); (2)utilized animal models; (3) had inadequate original data or substantial missing key data; (4) The treatment group excluded literature on combining 5-ASA with other drugs.

## Outcome measures

**Primary outcome measures—Symptom relief.** (1) The clinical efficacy rate was an essential metric for assessing the overall effectiveness of BBR in treating UC [15]. The clinical efficacy rate was assessed against the following benchmarks for therapeutic efficacy ① Complete remission, defined as a bowel movement frequency of three or fewer times per day, absence of bloody stools, abdominal pain, and diarrhea, coupled with colonoscopic evidence of mostly normal mucosa. ② Effective treatment is marked by pronounced improvement in clinical symptoms, a decreased presence of blood in stools, and mild mucosal inflammation or the appearance of pseudopolyps, as shown by colonoscopic examinations. ③ Ineffective treatment, defined as the absence of any discernible improvement in clinical symptoms, colonoscopy results, or pathological findings. The formula for calculating Total effectiveness is Total effectiveness = Number of complete remission + Number of effective cases / Total number of cases * 100% [16].

**Secondary outcome measures—Serological indicators and safety profiles.**

1. Baron score: Evaluates the intestinal mucosal damage in patients with active UC [17]. The higher the Baron score, the more severe the colonic mucosal damage [18].

2. DAI: The average rectal bleeding score, stool frequency, and percentage weight change. A higher DAI score indicates more severe UC [19].

3. Symptom remission status: The duration over which patients experienced relief from abdominal pain, diarrhea, and pus-involving bloody stools [20].

4. Inflammatory cytokines: IL-6, IL-8, TNF-α, IL-10, IL-6, IL-8, and TNF-α levels decrease while IL-10 increases, indicating a reduction in UC's inflammatory response [21].

5. Immune response: CD4+ T cells, CD8+ T cells, CD4+/CD8+ ratio. CD4+ and CD8+ T lymphocytes are distinct subpopulations that play pivotal roles in the immune response, with the CD4+/CD8+ ratio serving as a gauge of immune function in UC patients [22].

6. Adverse reactions.

### Literature screening, data extraction, and quality assessment

By screening the titles and abstracts, two authors (Jilei Li and Chenchen Zhang) independently removed duplicate and irrelevant records. Any research discrepancies were addressed by consulting the third researcher, Yanchao Xu. After skimming complete text, articles not meeting the inclusion criteria or meeting the exclusion criteria were discarded. Two independent researchers summarized the extracted literature information into a table following standardized instructions. Data extraction included the first author, publication year, sample size, age, sex ratio, interventions, treatment method, treatment time, outcome, and source. If the data in the table were missing, they were not included. The assessment of the quality of the included studies was meticulously conducted following the Cochrane Handbook for Systematic Reviews of Interventions, version 5.4.1 [23], which provided a risk of bias-evaluation tool. We used version 2.0 of the Cochrane Risk-of-Bias (RoB) instrument for risk-of-bias assessment.

### Statistical analysis

Our statistical analyses were performed with RevMan, which was tailored explicitly for conducting meta-analyses. Dichotomous variables were evaluated using relative risks (RRs), and continuous variables were assessed using the standard mean difference (SMD), both with accompanying 95% CI. In instances where heterogeneity tests yielded an $I^2 < 50\%$, indicating no significant heterogeneity among the groups, a fixed-effects model was applied. In contrast, a random effects model was adopted where significant heterogeneity ($I^2 > 50\%$) was detected, with $P < 0.05$ indicating statistical significance. The potential for publication bias was examined through the construction of funnel plots.

## Results

### Search results

The preliminary literature search retrieved a total of 1235 publications, distributed as follows: Medline/PubMed(169), Sinomed (77), Embase (138), CNKI (401), Wanfang (198), and VIP (152). An exhaustive screening process led to the inclusion of 10 RCTs in our study. The detailed flow of literature selection and the results thereof are depicted in Fig 1.

### Characteristics of the included studies

Our meta-analysis included 10 RCTs [24–33] enrolling 952 UC patients who were evenly divided into 476 in the treatment cohort and 476 in the control cohort. We meticulously extracted patient characteristics such as the first author, publication year, sample size, age, sex ratio, interventions, treatment method, treatment time, outcome, and source. These fundamental characteristics are delineated in Table 1.

### Quality assessment of the included studies

The quality evaluation of the selected studies included an examination of seven specific criteria, namely, selection bias, performance bias, detection bias, attrition bias, reporting bias, and

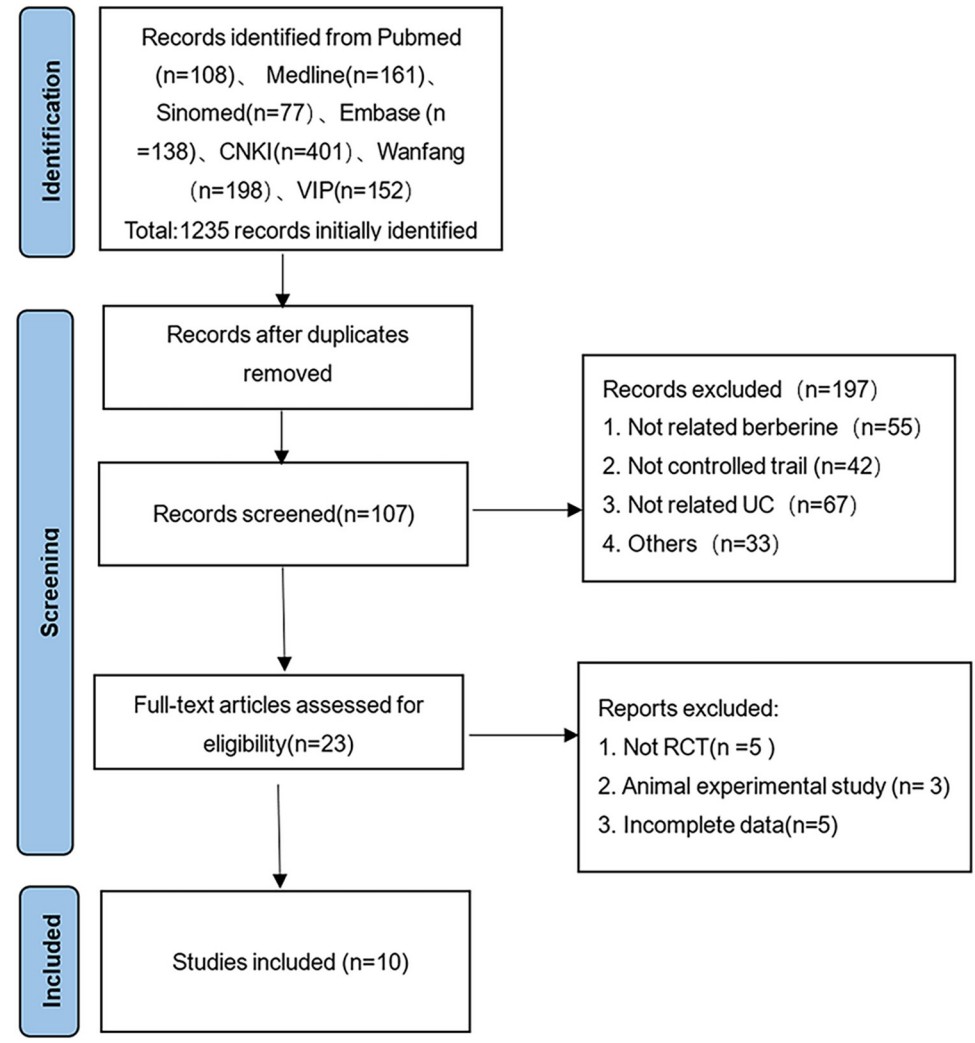

**Fig 1. PRISMA flowchart of the study selection process.**

any other possible sources of bias. Among them, four studies [27, 28, 30, 32] detailed the use of random number tables to generate random sequences, while the others mentioned the use of randomization without elaboration. Notably, no explicit descriptions of the allocation concealment methods used were provided. The specifics of the quality assessment are graphically represented in Fig 2. Bias risk assessment for individual studies is shown in S1 Fig.

## Meta-analysis outcomes

**Symptom relief—Clinical efficacy rate.** In our meta-analysis, 8 articles [24–27, 29–31, 33], with a total of 778 patients, reported this rate. The studies were subjected to a fixed-effects model analysis because of the low heterogeneity ($I^2 = 0\%$, P < 0.99). BBR significantly enhanced the clinical efficacy of 5-ASA in managing UC compared with that of monotherapy (RR = 1.22, 95% CI [1.15, 1.30], $P < 0.00001$), as depicted in Fig 3.

**Serological indicators.**

*(1) Baron score*. Our research included data from 2 studies [30, 31] that included a cohort of 186 subjects and pre-provided and posttreatment Baron score. The results indicated

**Table 1. Characteristics of studies included in the meta-analysis.**

| ID | Sample Size | Age(E/C)mean or mean±SD | (Male/Female) (E/C) | Interventions(E) | Interventions (C) | Treatment Method | Treat Time /W | Outcome | Source |
|---|---|---|---|---|---|---|---|---|---|
| Chen 2020 [24] | 37/37 | E: 33.87±3.42 | E: 24/13 | BBR: 0.2 g, tid | 5-ASA | Oral | 12 | ①⑥⑦⑧⑨ | CNKI |
| | | C: 34.02±3.44 | C: (-) | 5-ASA: 0.5 g, tid | 0.5–1.0 g, tid | | | | |
| Chen 2021 [25] | 58/58 | E: 35.7±9.60 | E: 35/23 | BBR: 2.0 g, tid | 5-ASA | Enema | 3 | ①②④⑨⑪ | CNKI |
| | | C: 35.2±9.80 | C: 33/25 | 5-ASA: 2.0 g/d | 0.5 g, qid | | | | |
| Cheng 2022 [26] | 68/68 | E: 40.15±11.37 | E: 39/29 | BBR: 1.0 g, qid | 5-ASA | Enema | 8 | ①⑥⑧⑨ | CNKI |
| | | C: 39.55±1.26 | C: 37/31 | 5-ASA: 0.2 g, tid | 1.0 g, qid | | | | |
| Cui 2021 [27] | 47/47 | E: 41.37±1.42 | E: 24/23 | BBR: 0.3 g, tid; | 5-ASA | Oral | 12 | ①②③④⑦ | CNKI |
| | | C: 41.01±1.13 | C: 24/23 | 5-ASA: 1–2 g, tid | 1.0 g, qid | | | | |
| Gan 2020 [28] | 32/32 | E: 44.17±7.64 | E: 18/14 | BBR: 0.2g, tid | 5-ASA | Oral | 12 | ①②③⑤⑦ | CNKI |
| | | C: 44.53±8.29 | C: 17/15 | 5-ASA: 4-6g/d | 4–6 g/d | | | | |
| Wang 2020 [29] | 46/46 | E: 39.77±6.49 | E: 22/24 | BBR: 0.2 g, tid | 5-ASA | Oral | 8 | ①⑥⑨⑪ | CNKI |
| | | C: 39.28±6.13 | C: 25/21 | 5-ASA: 1.0 g, qid | 1.0 g, qid | | | | |
| Wang 2021 [30] | 51/51 | E: 47.52±5.36 | E: 27/24 | BBR: 0.2g, tid, | 5-ASA | Oral | 8 | ①⑤⑥⑦⑨⑩ | CNKI |
| | | C: 45.96±5.27 | C: 28/23 | 5-ASA: 0.5–1 g, tid | 0.5–1 g, tid | | | | |
| Wei 2019 [31] | 42/42 | E: 52.31±3.17 | E: 23/19 | BBR: 0.3 g, tid | 5-ASA 1.0 g, tid | Oral | 4 | ①②③④⑦⑨⑩ | CNKI |
| | | C: 49.48±5.82 | C: 23/19 | 5-ASA: 1 g, tid | | | | | |
| Xu 2020 [32] | 55/55 | E: 40.23±7.45 | E: 31/24 | BBR: 0.2 g, tid | 5-ASA | Oral | 8 | ①②③④⑤⑥⑦⑧⑨ | VIP |
| | | C: 36.86±5.32 | C: 32/23 | 5-ASA: 1 g, tid | 1 g, tid | | | | |
| Zhu 2018 [33] | 40/40 | E: 37.87±4.12 | E: 23/17 | BBR: 0.2 g, tid | 5-ASA | Oral | 8 | ①②③④⑥⑦⑧⑨ | CNKI |
| | | C: 34.62±3.44 | C: 22/18 | 5-ASA: 1 g, tid | 1 g, tid | | | | |

E: Experimental C: Control BBR: Berberine g: Gram W: Week 5-ASA: Mesalazine/Sulfasalazine

① Total effective rate ② Time of loss of abdominal pain ③ Time of loss of pus and blood stool ④ Time of loss of Diarrhea ⑤ DAI ⑥ IL-6 ⑦ IL-8 ⑧ IL-10 ⑨ TNF-α ⑩ Baron ⑪ Adverse reactions

substantial heterogeneity ($I^2 = 66\%$, $P = 0.09$). Employing a random-effects model to analyze the data revealed that patients receiving BBR plus 5-ASA had significantly greater Baron score than those treated with 5-ASA. The statistical relevance of these improvements

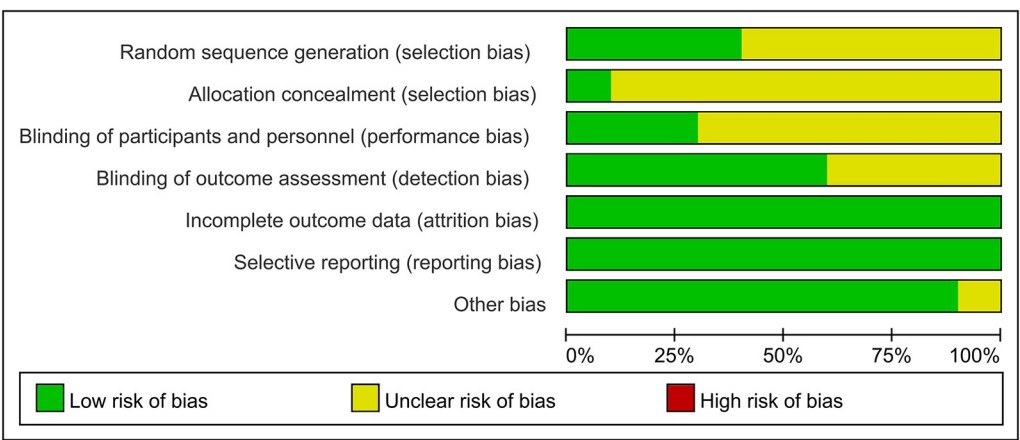

**Fig 2. Risk of bias summary.**

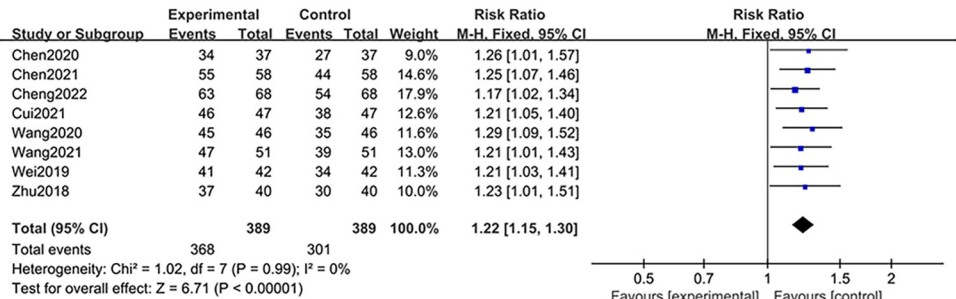

**Fig 3. Meta-analysis of clinical effective rate.**

was confirmed by a marked difference (SMD = -1.72, 95% CI [-2.30, -1.13], $P$ < 0.00001), as illustrated in Fig 4.

*(2) DAI assessment*. Analysis of data from 4 studies [27, 30, 32] encompassing a sample of 390 patients revealed a high level of heterogeneity ($I^2$ = 96%, $P$ < 0.000001). The application of a random effects model demonstrated that compared with treatment with 5-ASA, BBR plus 5-ASA therapy significantly reduced the DAI score. This difference was statistically significant, as highlighted by the standardized mean difference (SMD = -2.93, 95% CI [-4.42, -1.43], $P$ < 0.00001), as illustrated in Fig 5.

*(3) Symptom amelioration analysis*. 4 studies [25, 27, 32, 33] comprising 285 patients reported the duration of abdominal pain relief, showing a significant degree of heterogeneity ($I^2$ = 95%, $P$ < 0.00001). A random-effects model analysis revealed that BBR significantly shortened the duration of abdominal pain ($P$ < 0.00001). Similarly, 4 studies [25, 27, 32, 33] documented the duration of diarrhea relief in a cohort of 285 patients, showing substantial heterogeneity ($I^2$ = 96%, $P$ < 0.00001). The results indicated that BBR significantly reduced the duration of diarrhea ($P$ < 0.00001). Another set of 4 studies [25, 27, 32, 33] discussing the duration of hematochezia relief in 285 patients indicated high heterogeneity ($I^2$ = 97%, $P$ < 0.00001), with the random-effects model demonstrating that BBR meaningfully shortened the duration of relief ($P$ = 0.0004). These findings suggest that BBR can effectively adjunctive treat UC by attenuating symptoms such as abdominal pain, diarrhea, and hematochezia (SMD = -2.74, 95% CI [-3.45, 2.02], $P$ < 0.00001) as illustrated in Fig 6.

*(4) Inflammatory cytokines*. 6 studies [24, 26, 28, 30, 32, 33] assessing the serum IL-6 concentration in a cohort of 328 patients revealed substantial heterogeneity ($I^2$ = 67%, $P$ = 0.009). Using a random-effects model, the study revealed that treatment with BBR resulted in a

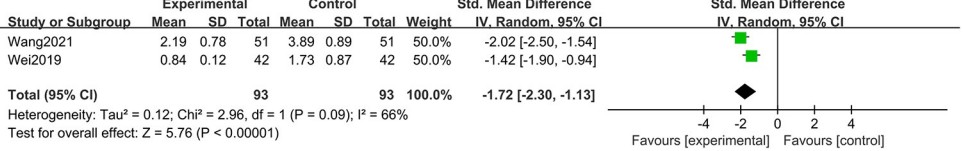

**Fig 4. Meta-analysis of Baron score.**

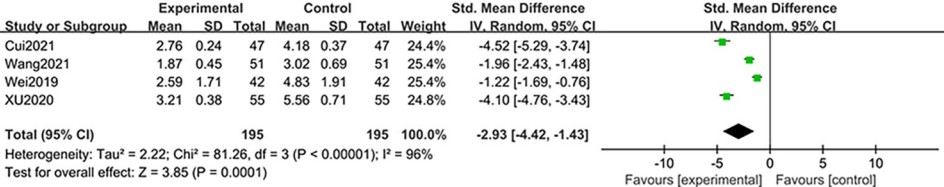

**Fig 5. Meta-analysis of DAI score.**

significant decrease in IL-6 expression, which was statistically significant ($P < 0.00001$). 7 studies [24, 26, 28, 30–33] reported on IL-8 levels across 366 participants, indicating pronounced heterogeneity ($I^2 = 83\%$, $P < 0.00001$). The results of these studies confirmed that BBR significantly inhibited the expression of IL-8, a notable proinflammatory mediator ($P < 0.00001$). In addition, 3 studies [24, 32, 33] focusing on IL-10 levels in 264 subjects showed minimal heterogeneity ($I^2 = 0\%$, $P = 1.00$). Unlike IL-6, IL-8, and TNF-α, IL-10 is a common anti-inflammatory factor. IL-10 is a common anti-inflammatory factor, and the higher its expression level, the milder the inflammatory response [34]. 3 studies [24, 32, 33] suggested that compared to the use of 5-ASA alone, the combination of BBR and 5-ASA can increase the expression of IL-10, helping to alleviate the inflammatory response in patients with UC. These studies suggest that BBR administration reduces the levels of the anti-inflammatory cytokine TNF-α in patients with UC ($P < 0.00001$). Finally, an examination of TNF-α concentrations, as reported by 8 studies [24, 26, 28, 30–33] involving 858 patients, revealed substantial heterogeneity ($I^2 = 84\%$, $P < 0.00001$). Subsequent analysis with the random effects model confirmed that BBR significantly lowered the serum TNF-α concentration, a finding of statistical and clinical relevance ($P < 0.00001$). These collective results underscore the efficacy of BBR in modulating inflammatory cytokines, thereby delineating its therapeutic potential in the management of UC; these findings are

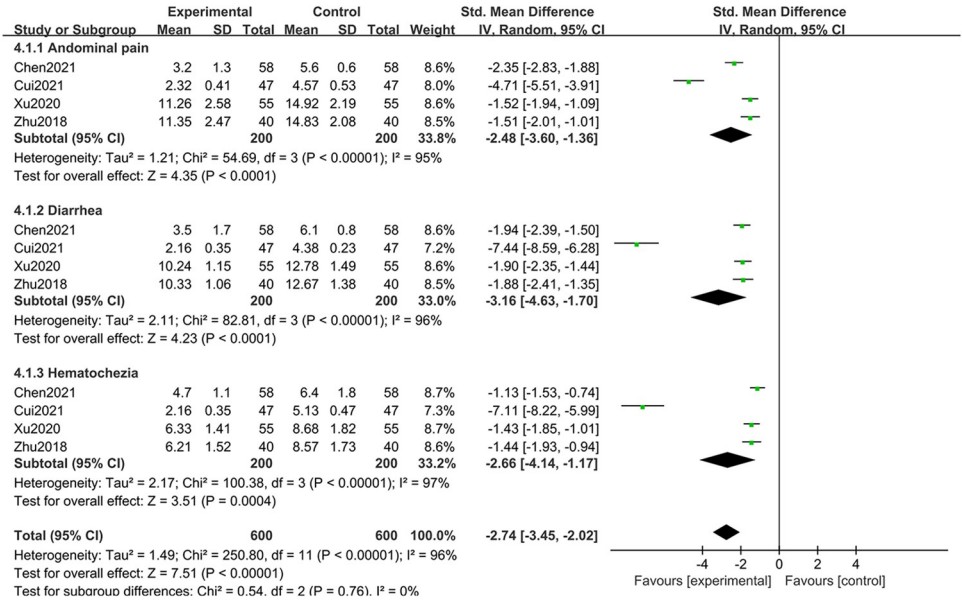

**Fig 6. Meta-analysis of clinical symptoms.**

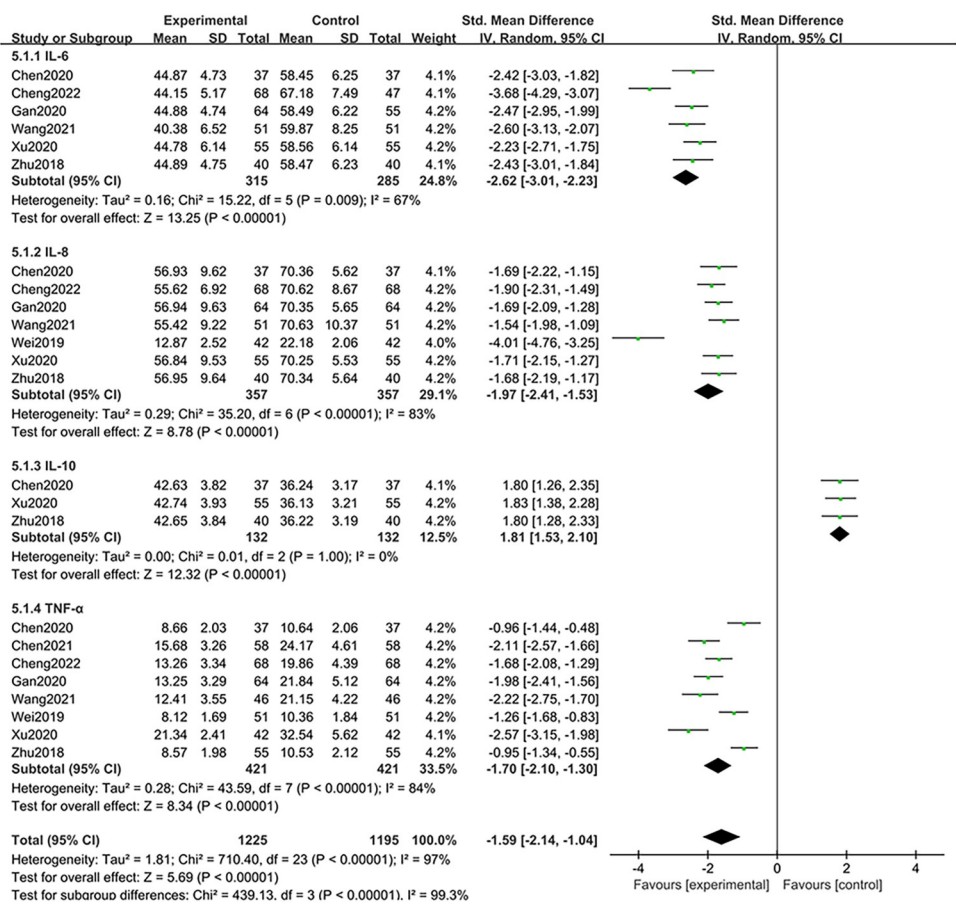

**Fig 7. Meta-analysis of inflammatory cytokines.**

corroborated by statistically significant data [SMD = -1.59, 95% CI [-2.14, -1.04], $P < 0.00001$), as depicted in Fig 7.

*(5) Immune response.* An analysis of 2 studies [26, 28] encompassing a sample of 264 patients demonstrated minimal heterogeneity ($I^2 = 0\%$, $P = 0.34$) and revealed that BBR administration resulted in a significant increase in CD4+ T cells ($P < 0.00001$). The increase in CD4 + T cells may promote an increase in the number of regulatory T cells, which helps alleviate the inflammatory response in UC. Similarly, 2 studies [26, 28] assessing CD8+ T cells in the same patient cohort reported low heterogeneity ($I^2 = 37\%$, $P = 0.21$). The data showed that the combination of BBR with 5-ASA did not significantly alter the percentage of CD8+ T cells compared to the effect of 5-ASA alone ($P = 0.50$). Moreover, statistics regarding the CD4+/CD8+ ratio from the studies above exhibited considerable heterogeneity ($I^2 = 81\%$, $P = 0.02$). Using a random effects model for analysis indicated a significant increase in the CD4+/CD8+ ratio, which suggested an improvement in immune function ($P = 0.0001$). These data suggest that BBR has the potential to favorably alter the immune response in the context of UC management. The therapeutic implications of these modifications are statistically significant and substantiated by the standardized mean difference [SMD = 1.06, 95% CI [0.24, 1.87], $P < 0.00001$], as depicted in Fig 8.

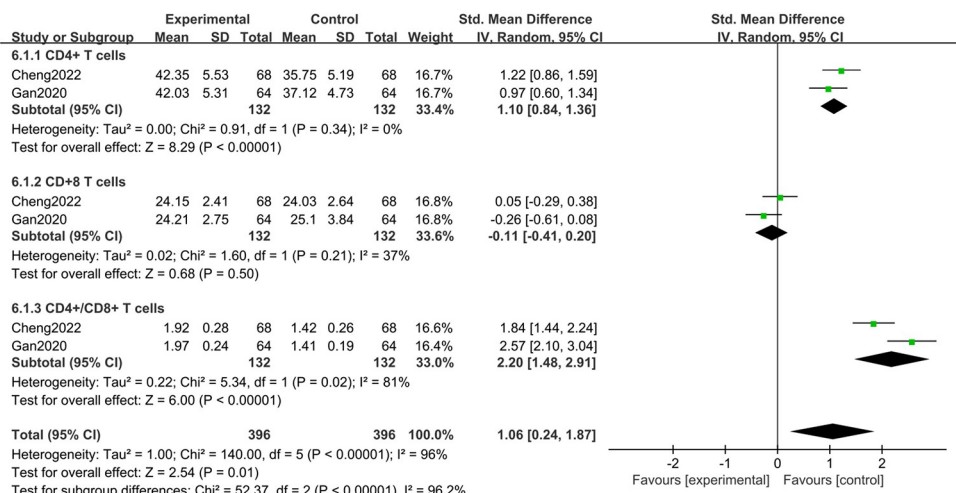

**Fig 8. Meta-analysis of CD4+ T cells, CD8+ T cells, CD4+/CD8+ ratio.**

**Safety profiles—Adverse reactions.** An analysis of four studies [24–26, 29] documented adverse effects, including nausea, vomiting, rash, and abdominal bloating, across a cohort of 418 patients. Within this cohort, adverse reactions occurred in 18 patients in the treatment group and 24 in the control group. The aggregated data reflected minimal heterogeneity ($I^2 =$ 0%, $P = 0.84$). An analysis conducted through a fixed-effects model indicated a lack of significant differences in the occurrence of adverse reactions between the BBR treatment group and the control group (RR = 0.75, 95% CI [0.42, 1.33]; $P = 0.32$), as illustrated in Fig 9.

**Sensitivity analysis.** A sensitivity analysis of the effectiveness of BBR in treating UC was performed by systematically excluding individual studies and reassessing the outcomes. The recalculated data showed no noteworthy differences from the original findings ($P > 0.05$), thereby confirming the robustness of the meta-analysis.

**Bias risk assessment.** The assessment of publication bias in the clinical efficacy of BBR for treating UC was conducted through funnel plot analysis. This analysis employed the RR of the consolidated studies on the x-axis against the inverse of their log (RR) on the y-axis. The statistical analysis suggested the presence of potential publication biases in the compiled research, as depicted in Fig 10.

## Discussion

UC is a complex gastrointestinal disease characterized by recurrent episodes and challenging management, profoundly affecting patients' quality of life [35]. BBR can be a safe and

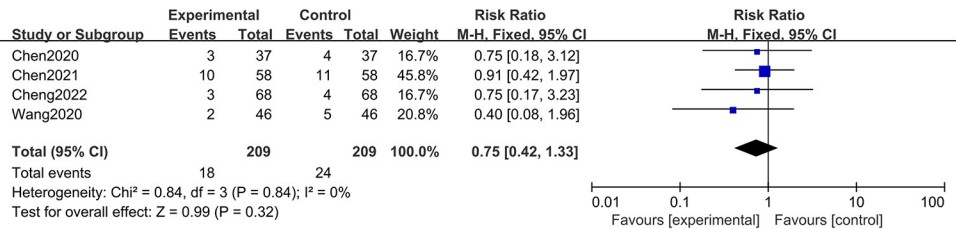

**Fig 9. Meta-analysis of adverse reactions.**

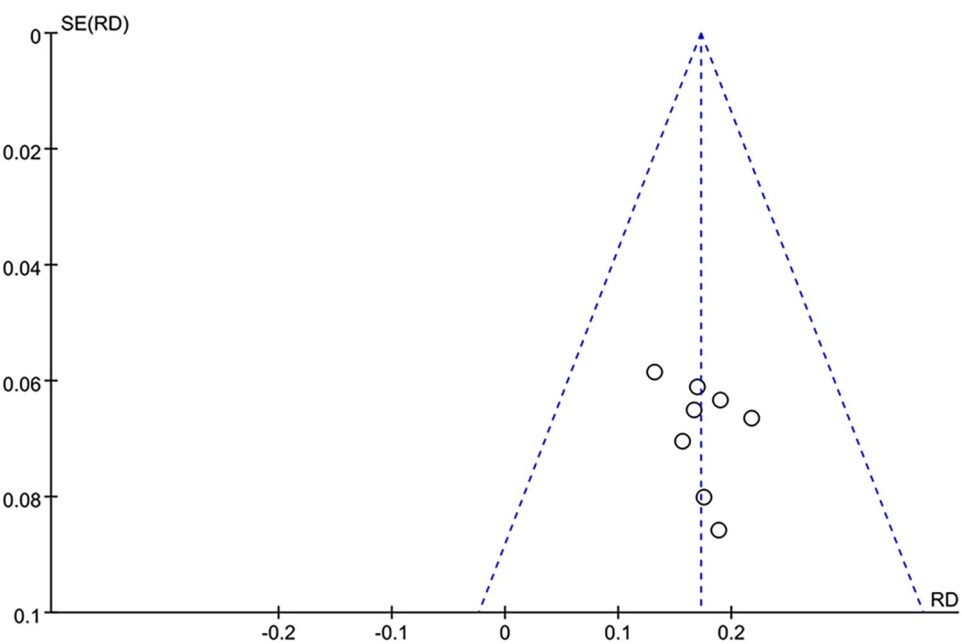

**Fig 10. Clinical effective rate funnel chart.**

productive complementary intervention to support UC disease management. It can potentially attenuate symptoms, including abdominal pain and diarrhea, improving patient outcomes [36]. Xiong et al. [37] demonstrated through mouse experiments that BBR can improve intestinal mucosal damage, promote mucosal healing, and alleviate symptoms such as diarrhea and bloody stools. Moreover, BBR facilitates the modulation of immune functions in UC mice by activating the IL-4/STAT6 signaling pathway, inhibiting M1 macrophages, and promoting M2 macrophages [38]. M1 and M2 macrophages can influence the differentiation of CD4+ and CD8+ T cells by producing factors like TNF-α and IL-10, thereby regulating the immune function in UC mice and suppressing excessive inflammatory responses [39]. Li et al. [40] research suggested that BBR can alleviate the inflammatory response in a cat model of UC, reduce inflammatory factors, and improve conditions of diarrhea and bloody stools. Li et al. further mechanistic studies showed that BBR can decrease *Bacteroidetes* and increase *Firmicutes* in DSS-treated cats. BBR reshaped the microbiota composition by reducing the abundance of *Proteobacteria*. Our study provides evidence suggesting that the application of BBR plus 5-ASA for treating UC may augment clinical efficacy, diminish Baron endoscopic score, decrease DAI score, ameliorate clinical manifestations, alleviate inflammatory processes, and adjust immune responses. Nonetheless, its role in mitigating adverse reactions is not significant.

Our analysis systematically addresses the data in three domains: symptom relief, serological indicators, and safety profiles. Focusing initially on symptom relief, prior investigations have indicated that BBR-containing traditional Chinese medicine formulations may significantly improve the clinical efficacy of UC management. Our extensive research, incorporating data from 778 subjects, robustly corroborates that BBR adjunct therapy substantially enhances clinical response rates among UC patients. Mucosal healing within the intestinal tract is a critical indicator of UC activity [41]. The Baron score is an indicator used to assess the severity of intestinal mucosal bleeding [42]. Our study also suggested that combining BBR with 5-ASA

treatment can reduce the Baron score in patients with UC, thereby improving the condition of the intestinal mucosa.

Converging evidence suggests that patients who achieve mucosal healing tend to exhibit higher rates of sustained remission and a diminished risk of relapse, a trend competently captured by the Baron endoscopic score—wherein a lower score signifies more favorable mucosal recovery [18].

The results of the endoscopic assessments in our study suggest that following BBR intervention, BBR is instrumental in lowering the Baron score, hence fostering mucosal restitution. Furthermore, BBR can reduce the DAI score and shorten the duration of symptoms such as diarrhea, rectal bleeding, and abdominal pain, indicating that BBR as a supplementary treatment has a specific efficacy in managing UC. Considering these outcomes, we postulate that there is a correlation between the reduction in patients' Baron score, therapeutic effectiveness, and symptom relief. BBR facilitates intestinal mucosal recovery, leading to a concomitant decrease in the Baron score; tangible symptom alleviation—specifically, in abdominal discomfort and diarrhea—is observed, culminating in discernible advances in UC treatment efficacy.

Emerging research has demonstrated a significant association between systemic inflammatory cytokine levels and the progression of UC. Increased concentrations of proinflammatory cytokines such as IL-6, IL-8, and TNF-$\alpha$ indicate heightened disease severity, whereas elevated IL-10 levels typically signal a mitigated disease state [43]. Research has shown that upon exposure to chemical or environmental triggers, intestinal epithelial cells prompt monocytes to secrete IL-8 [44]. This event initiates the activation of NF-$\kappa$B, culminating in the activation of macrophages, a process that initiates the sequential release of proinflammatory mediators, notably TNF-$\alpha$ and IL-6 [45]. Notably, the anti-inflammatory cytokine IL-10 counters this effect by restraining the transcription and secretion of inflammatory factors by activated monocytes [46]. Our meta-analysis evaluated multiple inflammatory cytokines, and the evidence suggested that BBR potentially mitigates UC symptoms by attenuating the proinflammatory mediators IL-6, IL-8, and TNF-$\alpha$ and concurrently increasing IL-10 expression. By doing so, it appears to modulate inflammatory homeostasis within the organism.

Disproportionate inflammatory processes derail the body's immune equilibrium, invigorating T lymphocytes. Naive CD4+ T cells and naive CD8+ T cells become activated and differentiate into effector cells and/or memory cells [47]. Notably, CD4+ T cells can activate and differentiate an immature effector/memory CD4+ T cells phenotype, executing immune responses corresponding to their specific cell phenotypes [48]. At the same time, the increase in CD4+ T cells can maintain intestinal immune balance, reducing gut damage and inflammation in UC patients through immune surveillance and promoting inflammatory repair mechanisms [49]. CD8+ T cells—widely recognized as cytotoxic T lymphocytes—can directly eliminate pathogen-infected cells [50]. Under normal circumstances, the CD4+/CD8+ ratio decreases between 1 and 2, indicating healthy immune function. An imbalance in the CD4+/CD8+ ratio could exacerbate the inflammatory condition in patients with UC [51]. In this context, our findings revealed that BBR can favorably modulate the levels of CD4+ T cells, CD8+ T cells, and CD4+/CD8+ ratio, thereby improving immunological functions. The enhancement of immune functions not only includes the strengthening of immune responses in areas such as the gut mucosa, making it more effective in defending against pathogens but also in mitigating excessive inflammatory responses, thereby preventing damage to tissues [52]. In turn, this can alleviate the suffering of patients with UC. The study did not report the key immunological factors FOXP3 and CD25. FOXP3, as a primary marker of regulatory T cells, plays an essential role in the suppressive function of Tregs [53]. Research by Zhang et al. suggested that reduced expression of FOXP3 in the intestinal tissues of UC patients may lead to an imbalance in immune regulation, exacerbating pathological conditions and

inflammatory responses [54]. Similarly, a decrease in CD25 expression could weaken the suppressive capabilities effects of Tregs, further aggravating the pathological conditions and inflammation in UC [55].

The safety of BBR treatment for UC also warrants attention, with a focus on common symptoms such as nausea, abdominal pain, dyspepsia, constipation, and bloating. Our findings confirm the absence of significant adverse effects of BBR plus 5-ASA, confirming its favorable safety profile. However, this study has certain limitations. For example, all articles in the meta-analysis originated from China/Asia, which limits the generalizability of the results. Additionally, there is variability in the administration routes and treatment duration across different studies, which may limit the application of BBR as an adjunctive treatment.

## Conclusion

The research results reported in this article suggest that BBR plus 5-ASA for the treatment of UC is a safe and effective intervention. Nevertheless, the validity of these results is tempered by the constrained scope and caliber of the included studies, characterized by an absence of reporting on allocation concealment, blinding, and a scarcity of longitudinal follow-up. Therefore, interpreting the long-term therapeutic benefits of BBR with circumspection is imperative. This inquiry points to the promising role of BBR in improving UC but underlines the necessity for future research endeavors. There is a profound need for strategically designed, expansive, multi-institutional, high-caliber RCTs to consolidate and expand upon these preliminary findings.

## Supporting information

**S1 Checklist. PRISMA 2020 checklist.**
(DOCX)

**S1 Table. All Studies identified in the literature search.**
(XLSX)

**S2 Table. Raw data for meta-analysis.**
(XLSX)

**S1 Fig. Bias risk assessment for individual studies.**
(TIF)

## Acknowledgments

We would like to thank AJE (https://china.aje.com/cn) for English language editing.

## Author Contributions

**Conceptualization:** Jilei Li, Lili Yang.

**Data curation:** Jilei Li, Chenchen Zhang, Yanchao Xu.

**Investigation:** Chenchen Zhang, Lili Yang.

**Methodology:** Yanchao Xu.

**Supervision:** Chenchen Zhang, Yanchao Xu.

**Validation:** Jilei Li, Chenchen Zhang.

**Writing – original draft:** Jilei Li.

**Writing – review & editing:** Jilei Li, Chenchen Zhang, Lili Yang.

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
