## [Decision Letter · Decision Letter 0]

20 Feb 2024

PONE-D-24-02154Efficacy and safety of berberine for ulcerative colitis: A systematic review and meta-analysisPLOS ONE

Dear Dr. Yang,

Thank you for submitting your manuscript to PLOS ONE. After careful consideration, we feel that it has merit but does not fully meet PLOS ONE’s publication criteria as it currently stands. Therefore, we invite you to submit a revised version of the manuscript that addresses the points raised during the review process.

Please respond to each of the reviewer comments, including the detailed uploaded response provided by Reviewer 1. I concur that all of the criticisms provided by the reviewers are necessary and constructive. In addition to the reviewers' comments, I have the following points that should be addressed.

First, I must emphasize a couple of important issues raised in the review process noticed by multiple parties including the reviewers and myself: (1) that some of the subject studies are not in English is a barrier to many other researchers; (2) overstatement of claims about use context of berberine (BBR). The latter issue can be addressed by carefully moderating the language of the manuscript (including title) to indicate the actual use of BBR here as a complementary therapeutic. The former issue can be addressed in a variety of ways, one which will improve PRISMA reporting. You note that data, code, and other materials are not available in the PRISMA checklist, which is not best practice for PLOS or principles of open science broadly. Providing such information from all studies and translated (with publisher permission) from the non-English studies could ameliorate both the Chinese language and PRISMA issues.

Additionally, I have the following questions comments that should be addressed:

Regarding the section on adverse events, did any of the studies report glycemic control indicators (glucose, insulin, HOMA-IR, etc.) or changes in body mass/weight?I think that you need to carefully re-visit the T cell section. First, the claim that BBR “augmented” CD8+ is not supported by simple visual inspection of the data; further even if some statistical test alleges significant difference here, it is not supported by the actual numbers/biology whose differences are not truly substantive.   Second, CD4+ changes could include changes in T regulatory (Treg) cell populations. Did any studies report FOXP3 and/or CD25 expression? This should be examined and discussed, even if none of the reported studies indicated such. These are important cells to consider in pathologies such as UC. Finally, what do increased CD4+ mean in the context of “favours control” as depicted in the figure? The statements and the figure do not seem to be in agreement. Without further details as to the phenotypes of CD4+ T cells, once cannot really say much about favoring one group vs. another; for example, an increase in Th17 cells could actually indicate worsening of cellular pathology, where as an increase in Treg may indicate amelioration.Line 120: be more detailed as to what is meant by “statistical inaccuracies”.Lines 148-152 need to be re-written to indicate the subject studies used these techniques. As currently written the statements could be construed that you conducted these experiments.Table 1 descriptors need to be re-written in English.I suggest combining Figures 3, 4, and 5 into a single figure.Lines 256-257: more should be stated about the IL-10 results, particularly discordance between control and BBR intervention.Delete the first sentence of line 340. It is unnecessary.There are currently no detailed figure legends. This should be addressed. Please submit your revised manuscript by Apr 01 2024 11:59PM. If you will need more time than this to complete your revisions, please reply to this message or contact the journal office at plosone@plos.org. Please include the following items when submitting your revised manuscript:A rebuttal letter that responds to each point raised by the academic editor and reviewer(s). You should upload this letter as a separate file labeled 'Response to Reviewers'.A marked-up copy of your manuscript that highlights changes made to the original version. You should upload this as a separate file labeled 'Revised Manuscript with Track Changes'.An unmarked version of your revised paper without tracked changes. You should upload this as a separate file labeled 'Manuscript'.

We look forward to receiving your revised manuscript.

Sincerely,

Nicholas A. Pullen, Ph.D.

Academic Editor

PLOS ONE

Journal Requirements:

3. Please include your tables as part of your main manuscript and remove the individual files. Please note that supplementary tables (should remain/ be uploaded) as separate "supporting information" files.

Additional Editor Comments (if provided):

Dear Dr. Yang,

Please respond to each of the reviewer comments, including the detailed uploaded response provided by Reviewer 1. I concur that all of the criticisms provided by the reviewers are necessary and constructive. In addition to the reviewers' comments, I have the following points that should be addressed.

First, I must emphasize a couple of important issues raised in the review process noticed by multiple parties including the reviewers and myself: (1) that some of the subject studies are not in English is a barrier to many other researchers; (2) overstatement of claims about use context of berberine (BBR). The latter issue can be addressed by carefully moderating the language of the manuscript (including title) to indicate the actual use of BBR here as a complementary therapeutic. The former issue can be addressed in a variety of ways, one which will improve PRISMA reporting. You note that data, code, and other materials are not available in the PRISMA checklist, which is not best practice for PLOS or principles of open science broadly. Providing such information from all studies and translated (with publisher permission) from the non-English studies could ameliorate both the Chinese language and PRISMA issues.

Additionally, I have the following questions comments that should be addressed:

• Regarding the section on adverse events, did any of the studies report glycemic control indicators (glucose, insulin, HOMA-IR, etc.) or changes in body mass/weight?

• I think that you need to carefully re-visit the T cell section. First, the claim that BBR “augmented” CD8+ is not supported by simple visual inspection of the data; further even if some statistical test alleges significant difference here, it is not supported by the actual numbers/biology whose differences are not truly substantive. Second, CD4+ changes could include changes in T regulatory (Treg) cell populations. Did any studies report FOXP3 and/or CD25 expression? This should be examined and discussed, even if none of the reported studies indicated such. These are important cells to consider in pathologies such as UC. Finally, what do increased CD4+ mean in the context of “favours control” as depicted in the figure? The statements and the figure do not seem to be in agreement. Without further details as to the phenotypes of CD4+ T cells, once cannot really say much about favoring one group vs. another; for example, an increase in Th17 cells could actually indicate worsening of cellular pathology, where as an increase in Treg may indicate amelioration.

• Line 120: be more detailed as to what is meant by “statistical inaccuracies”.

• Lines 148-152 need to be re-written to indicate the subject studies used these techniques. As currently written the statements could be construed that you conducted these experiments.

• Table 1 descriptors need to be re-written in English.

• I suggest combining Figures 3, 4, and 5 into a single figure.

• Lines 256-257: more should be stated about the IL-10 results, particularly discordance between control and BBR intervention.

• Delete the first sentence of line 340. It is unnecessary.

• There are currently no detailed figure legends. This should be addressed.

Reviewers' comments:

Reviewer's Responses to Questions

**Comments to the Author**

1. Is the manuscript technically sound, and do the data support the conclusions?

Reviewer #1: Partly

Reviewer #2: Yes

2. Has the statistical analysis been performed appropriately and rigorously? 

Reviewer #1: I Don't Know

Reviewer #2: Yes

3. Have the authors made all data underlying the findings in their manuscript fully available?

Reviewer #1: No

Reviewer #2: No

4. Is the manuscript presented in an intelligible fashion and written in standard English?

Reviewer #1: Yes

Reviewer #2: Yes

5. Review Comments to the Author

Reviewer #1: There is an attached review file with detailed comments by line/section of the article. However, below are a few general comments:

Overall, I think this analysis is of interest to the field and a good contribution to the literature. However, there are a few overarching concerns (“General Comments”) that must be addressed before I can recommend for the publication, in addition to a detailed breakdown of items associated with each section of the manuscript (“Detailed Comments”) in the attached review file.

GENERAL COMMENTS (again, PLEASE see detailed comments in the attached review file).

1. There are many locations where the authors overstate the findings/context of the study to implicate “BBR as a therapy for UC” when the study actually evaluates the efficacy of “BBR as a complementary therapy to support/enhance standard treatments.” Because BBR is being used in addition to standard treatment in all studies and is not being used alone in comparison to standard treatment, you are not evaluating BBR as an alternative therapy. So, in locations where it is implied the study is evaluating BBR in "treating disease,” it should be reworded to something like, "supporting disease management". The locations of these overstatements in the manuscript are highlighted in detail in “Detailed Comments.”

2. Many issues with citations: There is a considerable lack of citations for physiological and mechanistic statements throughout the article. These are highlighted in detail in “Detailed Comments.” Also, two of the ten articles included in the analysis are not properly cited in the References (in-text Figures include different author names for references 22 and 24)

3. Need additional details in Methods for reproducibility (These are highlighted in detail in “Detailed Comments.”), especially around data extraction and additional clarity around the inclusion criteria.

4. The lack of availability of many of these articles in the English language and/or on many databases common to Europe and the Americas presents a major barrier to researchers outside of China/Asia who may want to follow up on these articles. I unsuccessfully tried to search for many of these articles through Medline, EMBASE, PubMed, Google Scholar, and three separate university libraries for institutions that I am affiliated with (one of them being a major medical school library)—so, I was unable to confirm many methods or double-check measures. Because of this, authors must include which database they accessed each article from and any identifiable database numbers (e.g., DOI) in Table 1 to facilitate the reproducibility of the search and access to these papers.

5.In the introduction, authors must place the current review/meta-analysis in the context of any existing systematic reviews and meta-analysis on similar content (which do exist)—Describe what content those reviews/meta-analysis include, and how this current meta-analysis is different and contributes to the literature.

6. Highlight limitations in discussion or conclusion

Reviewer #2: 1. Authors should provide more detailed information on the sources of BBR in each study.

2. It is recommended to include the UC pathological staging of patients before treatment in the study.

3. Please calibrate the reference format in the manuscript to maintain consistency.

4. The title is "Efficacy and safety of berberine for ulcerative colitis: A systematic review and meta-analysis", but the majority of the data included in the meta-analysis is related to the combined treatment of BBR and 5-ASA. How to demonstrate the efficacy and safety of BBR through the use of BBR in combination with 5-ASA data? Please use concise and succinct language to describe your research findings regarding this matter.

6. PLOS authors have the option to publish the peer review history of their article (what does this mean?). If published, this will include your full peer review and any attached files.

Reviewer #1: No

Reviewer #2: No

---

## [Author Response · Author response to Decision Letter 0]

4 Jun 2024

Dear Editors and Reviewers:

We gratefully appreciate the editors and all reviewers for their time spent making positive and constructive comments. These comments are all valuable and helpful for revising and improving our manuscript entitled “Efficacy and safety of berberine plus 5-ASA for ulcerative colitis: A systematic review and meta-analysis” (PONE-D-24-02154).

We have made our utmost efforts to improve the manuscript in the following ways: (1) avoiding overestimating the effects of berberine and accurately describing its effects; (2) adding necessary citations; (3) supplementing explanations or rewriting parts that are difficult to understand. These revisions will help enhance the overall quality of the manuscript. The reviewer comments are laid out below in italicized font and specific concerns have been numbered. We have studied comments carefully and have made corrections which we hope meet with approval. Our response is given in normal font and changes/additions to the manuscript are given in red.

Thank you and best regards.

Yours sincerely,

Lili Yang

Kaifeng Central Hospital

Responses to Reviewers

Editor Comments

Comment 1: Some of the subject studies are not in English is a barrier to many other researchers.

Response 1: Thank you for your comment. We were sorry for our careless mistakes. We have already changed the non-English text below Table 1 to English text. (line 188-190 on page 10).

Comment 2: overstatement of claims about use context of berberine 

Response 2: We would like to express our sincere gratitude for your professional review of our article. Your feedback and suggestions were very helpful and insightful. You are such a patient and responsible reviewer, and I feel very honored to have your assistance. Based on your suggestion, we have made revisions. When describing the effects of berberine, we have rephrased our wording to avoid overstating the effects of berberine. For example, we have used phrases like "BBR as a safe and efficacious complementary intervention to support disease management", "BBR in conjunction with standard therapy to support the treatment of UC", "Berberine plus 5-ASA for ulcerative colitis ", "BBR demonstrates substantial efficacy in treating UC without causing severe adverse reactions and may serve as a viable complementary therapy” to accurately express the effects of berberine. 

(line 1 on page 1), (line 41-42 on page 2), (line 247 on page 12), (line 409-410 on page 18) et al.

Comment 3: Regarding the section on adverse events, did any of the studies report glycemic control indicators (glucose, insulin, HOMA-IR, etc.) or changes in body mass/weight?

Response 3: Thanks for your comment. We have carefully read the literature included this time, however, we did not find any studies that reported these indicators. The issue you mentioned is interesting, and we will focus on it in our future research. 

Comment 4: I think that you need to carefully re-visit the T cell section. First, the claim that BBR “augmented” CD8+ is not supported by simple visual inspection of the data; further even if some statistical test alleges significant difference here, it is not supported by the actual numbers/biology whose differences are not truly substantive. 

Response 4 Thank you for your professional and meticulous review; we apologize for our oversight. The expression "augmented CD8+" was incorrect. After reevaluating, we have revised the sentence in the manuscript from " The data indicated that BBR significantly augmented CD8+ T cells (P < 0.00001)" to "The data indicated that combining BBR with 5-ASA does not significantly alter the expression of CD8+ T cells compared to using 5-ASA alone." We thank you again for your expert guidance. (line 281-283 on page 14).

Comment 5: Second, CD4+ changes could include changes in T regulatory (Treg) cell populations. Did any studies report FOXP3 and/or CD25 expression? This should be examined and discussed, even if none of the reported studies indicated such. These are important cells to consider in pathologies such as UC. 

Response 5: Thank you for your professional review. While the original study did not report on the expression of FOXP3 and/or CD25, these elements are crucial. Following your advice, we have added related discussions in the manuscript. The specific additions are as follows: "Regrettably, the study did not report key immunological factors, FOXP3 and CD25. FOXP3, serving as the principal marker for Tregs, is critical for their suppressive function[1]. Research by Zhang et al. suggests that diminished expression of FOXP3 in the intestinal tissues of UC patients may lead to an imbalance in immune regulation, exacerbating the pathological conditions and inflammatory responses[2]. Similarly, reduced expression of CD25 can weaken the inhibitory capacity of Tregs, aggravating the pathology and inflammation associated with UC[3]." (line 391-398 on page 18).

Comment 6: Finally, what does increased CD4+ mean in the context of “favours control” as depicted in the figure? 

Response 6: Thank you for your guidance. We have not expressed ourselves before, but with your help, we can now articulate more precisely, as follows: "The increase in CD4+ T cells may promote an increase in the number of regulatory T cells, which helps alleviate the inflammatory response in UC." Additionally, we have made modifications to the manuscript accordingly. (line 278-280 on page 13 ).

Comment 7:The statements and the figure do not seem to be in agreement. 

Response 7:Thank you for your comment. We are sorry for our carelessness. We have rechecked and made specific modifications as follows: Firstly, change (I2 = 81%, P < 0.00001) into (I2 = 81%, P = 0.02). Change (P = 0.0004) into (P = 0.0001). Additionally, we have changed “The data indicated data that BBR significantly augmented CD8+ T cells (P < 0.00001).” to “The data showed that combining BBR with 5-ASA does not significantly alter the expression of CD8+ T cells compared to using 5-ASA alone(P = 0.50).” (line 282-284 on page 14).

Comment 8:Without further details as to the phenotypes of CD4+ T cells, one cannot really say much about favoring one group vs. another; for example, an increase in Th17 cells could actually indicate worsening of cellular pathology, whereas an increase in Treg may indicate amelioration.

Response 8:Thank you for your detailed comments and professional guidance. As you pointed out, CD4+ T cells include both T effector cells (such as Th1, Th2, Th17, etc.) and T regulatory cells, which are immunosuppressive. They have different physiological functions; for instance, an increase in Th17 cells may indicate the progression of UC, while an increase in Treg cells may suggest a mitigation of UC. We revisited the article and realized that the mentioned CD4+ T cell compartment includes both T effector cells and T regulatory cells. At the same time, we also provide the DOI or link of the articles in the references for easy access by the readers.

Comment 9: Lines 148-152 need to be re-written to indicate the subject studies used these techniques. As currently written the statements could be construed that you conducted these experiments.

Response 9:Thank you for your reminder; your guidance has been immensely helpful. You also mentioned in the following comment that this section was overly detailed and suggested we streamline it. The revised content is as follows:

(4) Inflammatory cytokines: IL-6, IL-8, TNF-α, IL-10. Levels of IL-6, IL-8, and TNF-α decrease, while IL-10 increases, indicating a reduction in the inflammatory response of UC.

(5) Immune response: CD4+ T cells, CD8+ T cells, CD4+/CD8+ ratio. CD4+ and CD8+ T lymphocytes are distinct subpopulations that play pivotal roles in the immune response, with the CD4+/CD8+ ratio serving as a gauge of immune function in UC patients. (6) Adverse reaction.(line 139-146 on page 6).

Comment 10 Table 1 descriptors need to be re-written in English.

Response 10hank you for your comment. We were sorry for our careless mistakes. We have already changed the non-English text below Table 1 to English text. (line 188-190 on page 10).

Comment 11 I suggest combining Figures 3, 4, and 5 into a single figure.

Response 11 Thank you for your suggestion. Figures 3, 4, and 5 represent the clinical efficacy rate, Baron score, and DAI score, respectively. Each represents different aspects, hence separating them into individual charts will make the results clearer. Therefore, we have decided not to combine them, but we equally appreciate your valuable suggestion. 

Comment 12 Lines 256-257: more should be stated about the IL-10 results, particularly discordance between control and BBR intervention.

Response 12Thank you for your guidance, we have added the following content.

IL-10 is a common anti-inflammatory factor, and the higher its expression level, the milder the inflammatory response. 3 studies suggest that compared to using 5-ASA alone, the combination of BBR and 5-ASA can increase the expression of IL-10, helping to alleviate the inflammatory response in patients with UC. (line 261-264 on page 13).

Comment 13 Delete the first sentence of line 340. It is unnecessary.

Response 13 Thank you for your professional guidance, we have already removed that sentence. 

Comment 14 There are currently no detailed figure legends. This should be addressed.

Response 14Thank you for the reminder. We have already placed each figure legend directly after the paragraph in which they are first mentioned. 

Journal Requirements:

Comment 15 Please ensure that your manuscript meets PLOS ONE's style requirements, including those for file naming. 

Response 15 Thank you for your guidance; we have revised the style of the manuscript according to the requirements, including those for file naming. 

Comment 16. Please include your tables as part of your main manuscript and remove the individual files. Please note that supplementary tables (should remain/ be uploaded) as separate "supporting information" files.

Response 16 Thank you for your meticulous guidance on formatting. We have placed Table 1 in the manuscript as per your instructions. (line 187-190 on page 9-10).

Reviewer #1:

GENERAL COMMENTS

Comment 1: There are many locations where the authors overstate the findings/context of the study to implicate “BBR as a therapy for UC” when the study actually evaluates the efficacy of “BBR as a complementary therapy to support/enhance standard treatments.” a. Because BBR is being used in addition to standard treatment in all studies and is not being used alone in comparison to standard treatment, you are not evaluating BBR as an alternative therapy. So, in locations where it is implied the study is evaluating BBR in "treating disease,” it should be reworded to something like, "supporting disease management".

Response 1: We would like to express our sincere gratitude for your professional review of our article. Your feedback and suggestions were very helpful and insightful. You are such a patient and responsible reviewer, and I feel very honored to have your assistance. Based on your suggestion, we have made revisions. When describing the effects of berberine, we have rephrased our wording to avoid overstating the effects of berberine. For example, we have used phrases like "BBR as a safe and efficacious complementary intervention to support disease management" and "BBR in conjunction with standard therapy to support the treatment of UC" to accurately express the effects of berberine. (line 1 on page 1), (line 41-42 on page 2), (line 247 on page 12), (line 409-410 on page 18)et al.

Comment 2: Many issues with citations: There is a considerable lack of citations for physiological and mechanistic statements throughout the article. These are highlighted in detail in “Detailed Comments.” Also, two of the ten articles included in the analysis are not properly cited in the References (in-text Figures include different author names for references 22 and 24)

Response 2: Thank you for reading our paper so carefully. We have added citations in the section on physiological and mechanistic statements. Additionally, we have made modifications to the original references 22 and 24 to ensure that the citations are more standardized. After our revisions, the article now cites approximately 20 more references than the previous version. A detailed response will be provided below the comments of the first reviewer. 

Comment 3: Need additional details in Methods for reproducibility (These are highlighted in detail in “Detailed Comments.”), especially around data extraction.

Response : Thank you for your valuable guidance on the manuscript. We have supplemented the methods as requested. The following paragraph is the revised content: 

"By screening the titles and abstracts, two authors (Jilei Li and Chenchen Zhang) independently removed duplicate and irrelevant records. Any discrepancies between the researchers were addressed by consulting the third researcher Yanchao Xu. After skimming the full text, articles not meeting the inclusion criteria or meeting the exclusion criteria were discarded. Two independent researchers summarized the extracted literature information into a table following standardized instructions. Data extraction includes the first author, publication year, sample size, age, sex ratio, interventions, treatment method, treatment time, outcome, and source. If the data in the table is missing, it will not be included. The assessment of the quality of the included studies was meticulously conducted following the Cochrane Handbook for Systematic Reviews of Interventions, version 5.4.1, which provided a risk of bias-evaluation tool. We used version 2.0 of the Cochrane Risk-of-Bias (RoB) instrument for risk of bias assessment.” (line 150-157 on page 7).

Comment 4: The lack of availability of many of these articles in the English language and on many databases common to Europe and the Americas presents a major barrier to researchers outside of China/Asia who may want to follow up on these articles. I unsuccessfully tried to search for many of these articles through Medline, EMBASE, PubMed, Google Scholar, and three separate university libraries for institutions that I am affiliated with (one of them being a major medical school library)—so, I was unable to confirm many methods or double-check measures. Because of this, authors must include which database they accessed each article from and any identifiable database numbers (e.g., DOI) in Table 1 to facilitate the reproducibility of the search and access to these papers.

Response 4: Thank you for your professional guidance. We apologize for any inconvenience our layout may have caused. We also appreciate your suggestions. However, placing the DOI or LINKs in Table 1 would require additional space and could detract from the table's appearance. Therefore, we have uploaded the DOIs or LINKs as an attachment, titled "DOI OR LINK." We have also listed the DOIs or LINKs below for your reference.

1. Chen 2020：DOI: 10.14164/j.cnki.cn11-5581/r.2020.10.063

LINK:https://kns.cnki.net/kcms2/article/abstract?v=yqeyU9EK6jT70Eh65eKmwK43ydBlFTIDStH0jexxpkVk6vtooH0y2vfgnSk_B8W9hkQO6HVgO9u3iA0BsjexoPSd_J--WD0t21BYlntP4Mda8D3jrgtTQzfcJnacYUHIrh04FGHrhFw=&uniplatform=NZKPT&language=CHS

2.Chen 2021 

LINK:https://kns.cnki.net/kcms2/article/abstract?v=yqeyU9EK6jR_XW5Z3mPDxkyme3NI4DoXhznzuWMqQmDji6RrU67zynGJ97EZJrLDmWYG5B7pv9e6TfxfRDraupGkVHZQ9070plbElSTDFNOcx1vR94oVQkXXzP0jGkDukRYC9DVIErU=&uniplatform=NZKPT&language=CHS

3. Cheng 2022 

LINK: https://kns.cnki.net/kcms2/article/abstract?v=yqeyU9EK6jT1dAmV7rL-KIpLPmpheHbLyT5QSnjXfiXeuX68BMDP33lIhHTW5AoN_Ax8JOa7EnH06ZVvB52FhVxsg2I1_OVecwp7dma3zYV1jEvkS_TugnUCjfAnZR7iTQ1XjkalIyo=&uniplatform=NZKPT&language=CHS

4. Cui 2021 

LINK: https://kns.cnki.net/kcms2/article/abstract?v=yqeyU9EK6jRX2Bo-wogWiT86QuBWDmzqStLSmOKGvbIBicLxqheF3Vwh3pv3eWHxGpcQA0ZIOwTG506uHe6sFLOF7K7AxeigNHTO8v0W5Qm2bKTFjq33aqLkCgzB9OJvbWfm0gMqUG8=&uniplatform=NZKPT&language=CHS

5. Gan 2020 doi：10.3969/j.issn.1671-038X.2020.12.10

LINK:https://kns.cnki.net/kcms2/article/abstract?v=yqeyU9EK6jRcsAtlmqq8ZFMv6NhdyrYXJ2X8

---

## [Decision Letter · Decision Letter 1]

9 Jul 2024

PONE-D-24-02154R1Efficacy and safety of berberine plus 5-ASA for ulcerative colitis : A systematic review and meta-analysisPLOS ONE

Dear Dr. Yang,

Thank you for submitting your manuscript to PLOS ONE. After careful consideration, we feel that it has merit but does not fully meet PLOS ONE’s publication criteria as it currently stands. Therefore, we invite you to submit a revised version of the manuscript that addresses the points raised during the review process. Thank you for your improved, revised submission. Substantial English language editing will still be required for me to endorse the paper further. Additionally, while you have figure titles, you still do not have figure legends. Please add these. I refer you to the PLOS page on this: https://journals.plos.org/plosone/s/figures. Specifically, from the page, under Figure legend tips: "Describe the key messages of a figure: provide a description of the figure that will allow readers to understand it without referring to the text."

We look forward to receiving your revised manuscript.

Sincerely,

Nicholas A. Pullen, Ph.D.

Academic Editor

PLOS ONE

Journal Requirements:

Additional Editor Comments:

Thank you for your improved, revised submission. Substantial English language editing will still be required for me to endorse the paper further. Additionally, while you have figure titles, you still do not have figure legends. Please add these. I refer you to the PLOS page on this: https://journals.plos.org/plosone/s/figures. Specifically, from the page, under Figure legend tips: "Describe the key messages of a figure: provide a description of the figure that will allow readers to understand it without referring to the text."

Reviewers' comments:

Reviewer's Responses to Questions

**Comments to the Author**

1. If the authors have adequately addressed your comments raised in a previous round of review and you feel that this manuscript is now acceptable for publication, you may indicate that here to bypass the “Comments to the Author” section, enter your conflict of interest statement in the “Confidential to Editor” section, and submit your "Accept" recommendation.

Reviewer #1: All comments have been addressed

Reviewer #2: All comments have been addressed

2. Is the manuscript technically sound, and do the data support the conclusions?

Reviewer #1: Yes

Reviewer #2: Yes

3. Has the statistical analysis been performed appropriately and rigorously? 

Reviewer #1: Yes

Reviewer #2: Yes

4. Have the authors made all data underlying the findings in their manuscript fully available?

Reviewer #1: Yes

Reviewer #2: Yes

5. Is the manuscript presented in an intelligible fashion and written in standard English?

Reviewer #1: No

Reviewer #2: Yes

6. Review Comments to the Author

Reviewer #1: The authors have made tremendous improvements to enhance the transparency and reproducibility of their methods, which now reflect the rigor required for a systematic review.

The authors have also thoroughly addressed previous reviewer comments and have improved the clarity and organization of their manuscript.

While the clarity of the manuscript is greatly improved, there are still several grammatical errors throughout. These include punctuation errors, issues with spacing (or lack thereof) between characters, missing prepositions or conjunctions, and incorrect numbering of lists (e.g., under Primary Outcome Measures there are two #1’s). Once these minor issues are fixed, I believe the article will be ready for publication.

Some examples are below, although more instances can be found in the manuscript:

-Under the Inclusion Criteria section-- No need to use colons (the “:” symbol) after symptomatic relief, serological indicators, etc., because the lists that follow are encapsulated in parentheses.

-Under the Exclusion Criteria section-- Line 117, there is a period where there should be a semi-colon.

-Under the Primary Outcome Measures section-- Check the numbering. Both “clinical efficacy rate” and “complete remission” are labeled with the number 1.

-Character spacing throughout the manuscript -- There are many locations where there is no space between characters that should have spaces (e.g., between a period and a new word, or between a colon and a new word, etc.).

-In the Discussion, line 379-- Currently “effector cells, memory cells”  “effector cells AND/OR memory cells.”

Reviewer #2: (No Response)

7. PLOS authors have the option to publish the peer review history of their article (what does this mean?). If published, this will include your full peer review and any attached files.

Reviewer #1: No

Reviewer #2: No

---

## [Author Response · Author response to Decision Letter 1]

1 Aug 2024

Dear Editors and Reviewers:

We gratefully appreciate the editors and all reviewers for their time spent making positive and constructive comments. These comments are all valuable and helpful for revising and improving our manuscript entitled “Efficacy and safety of berberine plus 5-ASA for ulcerative colitis: A systematic review and meta-analysis” (PONE-D-24-02154).

We have made our utmost efforts to improve the manuscript in the following ways: (1) Correct grammatical errors in the manuscript;(2) Review the manuscript’s formatting and simplify complex sentences for better readability;(3) Revise the manuscript according to the reviewers' comments, such as changing “effector cells, memory cells” to “effector cells and/or memory cells. ”

These revisions will help enhance the overall quality of the manuscript. The reviewer comments are laid out below in italicized font and specific concerns have been numbered. We have studied comments carefully and have made corrections which we hope meet with approval. Our response is given in normal font and changes/additions to the manuscript are given in red.

Thank you and best regards.

Yours sincerely,

Lili Yang

Kaifeng Central Hospital

Responses to Reviewers

Comment 1: Under the Inclusion Criteria section-- No need to use colons (the “:” symbol) after symptomatic relief, serological indicators, etc., because the lists that follow are encapsulated in parentheses.

Response 1: Thank you for your careful and professional guidance. We have removed the ":" in that section as you suggested. (line 107-109 on page 5)

Comment 2: Under the Exclusion Criteria section-- Line 117, there is a period where there should be a semi-colon.

Response 2: Thank you for reading our paper so thoroughly. We have made the necessary revisions. (line 115 on page 5)

Comment 3: In the Discussion, line 379-- Currently “effector cells, memory cells”  “effector cells AND/OR memory cells.”

Response 3:Thank you for your guidance. We have made the revisions according to your suggestions.

(line 388 on page 18)

Comment 4:Character spacing throughout the manuscript -- There are many locations where there is no space between characters that should have spaces (e.g., between a period and a new word, or between a colon and a new word, etc.).

Response 4: Thank you for your rigorous and specific review comments, which will enhance the readability of our manuscript. We also apologize for our errors. After a thorough review, we have corrected the incorrect character spacing in the manuscript. Here are a few examples of the revisions; more changes can be seen in the manuscript:

(1). We have added a space before the reference number in the citations. For example, changed “patients with active UC[17]” to “patients with active UC [17]”. (line 134 on page 6)

(2). Changed “(2)DAI” to “(2) DAI” (line 136 on page 6)

(3). Changed “Inflammatory cytokines: IL-6” to “Inflammatory cytokines: IL-6” (line 140 on page 6)

(4). Changed “BBR: 0.2g,tid” to “BBR: 0.2 g, tid”, “E:33.87±3.42” to “E: 33.87±3.42” (line 187 on page 10)

At the same time, we are also making revisions to other issues. Below are a few examples of the changes, with more modifications detailed in the manuscript.

(1). We have rephrased complex sentences. For instance, “Of particular interest is BBR, an efficacious alkaloid isolated from botanical sources such as Coptis chinensis and Phellodendron amurense, which was identified for its ability to relieve symptoms of abdominal discomfort and diarrhea within UC treatment paradigms” has been changed to “BBR, a potent alkaloid from Coptis chinensis and Phellodendron amurense, is noted for easing abdominal discomfort and diarrhea in UC treatment.” (line 62-63 on page 3)

(2). Standardized symbols: Changed “For the retrieval of literature, the search terms used were ‘ulcerative colitis’, ‘ulcerative’, ‘inflammatory bowel disease’, ‘BBR’, ‘BBR hydrochloride’, and ‘umbellate’ ‘randomized’ and ‘randomized controlled trial” has been changed to “For the retrieval of literature, the search terms used were ‘ulcerative colitis’, ‘ulcerative’, ‘inflammatory bowel disease’, ‘BBR’, ‘BBR hydrochloride’, ‘umbellate’, ‘randomized’, and ‘randomized controlled trial’.” (line 88-91 on page 4)

(3). Changed “in the treatment of UC” to “in treating UC”(line 121 on page 5)

(4). Changed “flow—chart” to “flowchart” (line 178 on page 8)

(5). Changed “w: Week” to “W: Week” (line 188 on page 10)

(6). Changed “I2” to “I2” (line 215 on page 11)

(7). Changed “（4）” to “(4) ” (line 256 on page 13)

(8). Changed “those with UC” to “UC patients” (line 353 on page 16)

(9). Changed “our” to “Our” (line 381 on page 17)

(10). Changed “Inflammatory” to “inflammatory” (line 382 on page 17)…

Comment 5: Under the Primary Outcome Measures section-- Check the numbering. Both “clinical efficacy rate” and “complete remission” are labeled with the number 1.

Response 5: Thank you for carefully reading our manuscript and providing valuable guidance. I believe we need explain the following: In the "Under the Primary Outcome Measures" section, (1) The evaluation of "clinical efficacy rate" includes three aspects, namely ①Complete remission; ②Effective treatment; and ③Ineffective treatment. Additionally, we have used different notations for "1" in the text. For example, we used (1) to annotate "clinical efficacy rate" and ① to annotate "complete remission". These two annotations are distinct. Based on the above situation, we have not made any changes here. Thank you again for your guidance. (line 120 on page 5)

Comment 6: Additionally, while you have figure titles, you still do not have figure legends. Please add these. 

Response 6: Thank you for your guidance and comments. We have added the figure legends as requested and marked them in red in the manuscript. The details are as follows.

Fig 1. PRISMA flowchart of the study selection process. A total of 10 studies meeting the requirements were included. (line 178 on page 8)

Fig 2. Risk of bias summary. The quality of the included studies was evaluated. (line 209 on page 11)

Fig 3. Meta-analysis of Clinical effective rate. Berberine adjunctive therapy can improve the clinical effective rate. (line 218 on page 11)

Fig 4. Meta-analysis of Baron score. Berberine adjunctive therapy can reduce the Baron score. (line 229 on page 12)

Fig 5. Meta-analysis of DAI score. Berberine adjunctive therapy can reduce the DAI score. (line 238 on page 12)

Fig 6. Meta-analysis of clinical symptoms. Berberine adjunctive therapy can alleviate symptoms of abdominal pain, diarrhea, and hematochezia. (line 253 on page 13)

Fig 7. Meta-analysis of inflammatory cytokines. Berberine adjunctive therapy can reduce levels of IL-6, IL-8, and TNF-α and increase IL-10 levels. (line 279 on page 13)

Fig 8. Meta-analysis of CD4+ T cells, CD8+ T cells, CD4+/CD8+ ratio. Berberine adjunctive therapy can increase levels of CD4+ T cells and the CD4+/CD8+ T cell ratio, with no significant effect on CD8+ T cell levels. (line 297 on page 14)

Fig 9. Meta-analysis of adverse reactions. Berberine adjunctive therapy has no significant adverse reactions. (line 309 on page 15)

Fig 10. Clinical effective rate funnel chart. The clinical effectiveness rate showed no significant bias risk. (line 324 on page 15)

---

## [Editor Report · Decision Letter 2]

7 Aug 2024

Efficacy and safety of berberine plus 5-ASA for ulcerative colitis : A systematic review and meta-analysis

PONE-D-24-02154R2

Dear Dr. Yang,

We’re pleased to inform you that your manuscript has been judged scientifically suitable for publication and will be formally accepted for publication once it meets all outstanding technical requirements.

Sincerely,

Nicholas A. Pullen, Ph.D.

Academic Editor

PLOS ONE

Additional Editor Comments (optional):

Thank you for your judicious attention to previous comments.
---

## [Editor Report · Acceptance letter]

27 Aug 2024

PONE-D-24-02154R2 

PLOS ONE

Dear Dr. Yang, 

I'm pleased to inform you that your manuscript has been deemed suitable for publication in PLOS ONE. Congratulations! Your manuscript is now being handed over to our production team.

Kind regards, 

on behalf of

Dr. Nicholas A. Pullen 

Academic Editor

PLOS ONE